

# $^{13}$C values of glycolytic amino acids as indicators of carbohydrate utilization in carnivorous fish

Yiming V. Wang[1], Alex H.L. Wan[2], Åshild Krogdahl[3], Mark Johnson[2] and Thomas Larsen[1]

[1] Department of Archaeology, Max Planck Institute for the Science of Human History, Jena, Germany
[2] Aquaculture Nutrition and Aquafeed Research Unit and Irish Seaweed Research Group, Ryan Institute, National University of Ireland, Galway, Ireland
[3] Department of Basic Sciences and Aquatic Medicine, Faculty of Veterinary Medicine, Norwegian University of Life Sciences, Oslo, Norway

Corresponding author
Yiming V. Wang, ywang@shh.mpg.de

## ABSTRACT

**Background**. Stable isotope analysis of single amino acids (AA) is usually applied in food web studies for tracing biosynthetic origins of AA carbon backbones and establishing trophic positions of consumers, but the method is also showing promise for characterizing quantity and quality of dietary lipids and carbohydrates.

**Methods**. To investigate whether changes in high- and low-digestible carbohydrates affect $\delta^{13}$C values of glycolytic AA, i.e., AA carbon backbones sourced from the glycolytic pathway, we compared Atlantic salmon (*Salmo salar*) from a feeding experiment with and without dietary inclusion of the red macroalga *Palmaria palmata*. The Control and experimental diets had similar relative proportions of macronutrients, but their ingredients differed; in the experimental treatment, 15% *Palmaria* inclusion substituted proteins from fishmeal and carbohydrates from corn starch.

**Results**. We found that $^{13}$C values of the glycolytic AA were highly sensitive to substitution of corn starch with *Palmaria*. The $\delta^{13}$C offsets of glycolytic AA between salmon and their diets were significantly greater in the *Palmaria* inclusion than Control treatment. This greater offset can be attributed to the different utilization of high- vs. low-digestible carbohydrate sources, i.e., corn starch vs. *Palmaria*, in the two treatments, and metabolic routing of dietary lipids. In addition, similar $\delta^{13}$C values of essential AA between treatments indicate similar nutrient assimilation efficiency for all terrestrial (pea protein concentrate and wheat gluten meal) and marine (fishmeal and red alga) derived protein sources. These results show that $\delta^{13}C_{AA}$ analysis is a promising tool for improving our understanding of how carnivorous fish utilize macronutrient and route metabolic intermediates to tissue.

## INTRODUCTION

Compound specific stable isotope analysis (CISA) of proteinogenic amino acids (AA) is an emerging tool for dietary reconstruction of finfishes. While stable carbon isotope analysis

of AA is predominantly applied for tracing biosynthetic origins of AA (*Larsen et al., 2009*; *Larsen et al., 2013*; *O'Brien, Fogel & Boggs, 2002*; *Scott et al., 2006*), a number of recent studies show that the method has the potential to characterize a consumer's intake of lipids and carbohydrates (*Leigh, Papastamatiou & German, 2018*; *Newsome et al., 2011*; *Newsome et al., 2014*; *Whiteman et al., 2018*). For example, CSIA has been applied to characterize assimilation and digestion of diets with varying macromolecular composition in e.g., tilapia and sharks, which in turn can be used to understand a species' nutritional requirements and ability to use resources in its environment (*Leigh, Papastamatiou & German, 2018*; *Newsome et al., 2011*; *Whiteman et al., 2018*). The origins of AA building blocks are highly diverse. Of the 20 proteinogenic AA, half of them are classified as nutritionally essential since animals cannot synthesize their carbon backbones, depending instead on essential EAA from the diet itself (*McMahon et al., 2010*; *Reeds, 2000*). This also means that the $\delta^{13}$C values of these AA typically match those in source protein with little or no isotopic offsets (*Jim et al., 2006*; *McMahon et al., 2010*; *O'Brien, Fogel & Boggs, 2002*). Some aquatic consumers may also rely on EAA supplemented from gut microbes but this pathway is usually confined to herbivores feeding on nutritionally insufficient diets (*Arthur et al., 2014*; *Newsome et al., 2011*). Metabolic routing of the other 10 proteinogenic AA, the non-essentials (NEAA), are much more complex than for the EAA because their carbon skeletons may either be incorporated directly into proteinogic tissue or synthesized *de novo* from metabolic intermediates. While the term non-essential implies that animals can synthesize them at a rate that meets the cellular demand for protein synthesis, it is well documented that adequate amounts of dietary NEAA are required for maximum growth and optimum health (*Horvath et al., 1996*; *Womack & Rose, 1947*). Thus, post-ingestive *de novo* synthesis of NEAA carbon skeletons may vary according to quality and supply of dietary proteins (*Newsome et al., 2011*).

Animals synthesize NEAA carbon backbones from glycolytic or Krebs cycle intermediates (*Berg et al., 2015*). The carbon intermediates for the glycolytic AA, glycine (Gly), serine (Ser), and alanine (Ala), are thought to derive predominantly from carbohydrates and lipids (*Fernandes, Nadeau & Grootes, 2012*; *Newsome et al., 2014*; *Wang et al., 2018*). For example, naturally occurring stable carbon isotope ($\delta^{13}$C) variations of Ala are correlated to human sugar consumption (*Choy et al., 2013*), and $\delta^{13}$C values of Gly and Ser are highly sensitive biomarkers of aquatic and terrestrial lipid origins in captive Atlantic salmon (*Wang et al., 2018*). In contrast, the Krebs cycle intermediate AA, aspartic acid (Asx), glutamic acid (Glx), and proline (Pro), are thought to be sourced predominantly from dietary proteins and lipids (*Fernandes, Nadeau & Grootes, 2012*; *Newsome et al., 2014*). How species synthesize NEAA from carbon intermediates depends on their digestive physiology and nutritional requirements. A study on carnivorous leopard sharks showed that a dietary shift from squid to tilapia did not affect their AA metabolism despite the diets' substantial differences in carbohydrate content (*Whiteman et al., 2018*). In contrast, a study with bonnethead shark showed that glycogen synthesized from dietary carbohydrates were used as intermediates for NEAA synthesis (*Leigh, Papastamatiou & German, 2018*). These different metabolic responses for sharks raise the question of how carnivorous bony fishes utilize dietary macronutrients, and whether we can use naturally occurring markers

for assessing metabolic responses to dietary changes. It is becoming increasingly important to understand how carnivorous fish species respond to dietary change because both wild and domesticated fishes are experiencing dramatic changes in the food or prey they eat. For example, Atlantic salmon is strictly carnivorous in the wild, but the proportion of plant-based ingredients in salmon aquafeed is approaching 80% (*Gatlin et al., 2007*). No other fish have gone through comparable changes in feeding ecology in the last decades, thus, it is important understanding the metabolic fate of the plant carbohydrate as well as their effect on fish health (*Hemre, Mommsen & Krogdahl, 2002*).

Carbohydrates are not essential for Atlantic salmon since they are obligate carnivores (*Krogdahl et al., 1999*), but they are a cheap source of energy for aquafeed production, and have useful pellet binding properties (*Hemre & Krogdahl, 1996*). While excessive inclusion of digestible carbohydrates in the feed may cause glycogen accumulation in the liver and impair the salmon's hepatic function (*Aksnes, 1995*; *Brudeseth, 1996*; *Frøystad et al., 2006*; *Hemre, Mommsen & Krogdahl, 2002*; *Tan et al., 2009*), no or very low carbohydrate inclusions will reduce protein retention (*Hemre et al., 1995*) in part because glucose formed from carbohydrates can divert AA away from oxidative pathways (*Cowey, De la Higuera & Adron, 1977*; *Sanchez-Muros et al., 1995*). The main carbohydrates sources in aquafeed for Atlantic salmon have, until now, derived from extruded maize, wheat and other plant-based starch in the forms of amylose and amylopectin (*Krogdahl, Sundby & Bakke, 2011*). These gelatinized starches are easier to digest compared to glucose and other mono- and polysaccharides (*Bogevik, 2015*). Furthermore, a high inclusion of complex carbohydrates may limit utilization of gelatinized starches (*Hemre, Mommsen & Krogdahl, 2002*). Despite these less desirable nutritional properties of complex carbohydrates, macroalgae meal is now under investigation as a potential feed additive to promote salmon health and digestion (*Øverland, Mydland & Skrede, 2017*), and already exists in commercial feed products including OceanFeed™ (Ocean Harvest, Gortnaloura, Ireland) and DigestSea® and Algimun® (Olmix, Brittany, France). In a scientific feeding trial with Atlantic salmon, the inclusion of the red marine intertidal macroalgae *Palmaria palmata*, also known as red dulse, significantly decreased alanine transaminase activity, which indicated an improved liver health (*Wan et al., 2016*). However, since *P. palmata* is rich in hardly digestible carbohydrates such as cellulose, hemicellulose and xylans (up to 60%) (*Jiao et al., 2012*), it is unclear how the inclusion of macroalgae affect carbohydrate utilization, lipid and carbohydrate interaction as well as AA synthesis in Atlantic salmon.

In this study we investigate how varying proportions of high- and low-digestible carbohydrates affect $\delta^{13}C_{NEAA}$ values by analyzing Atlantic salmon and their diets in a controlled feeding experiment (*Wan et al., 2016*). The two diets in our study have similar relative proportions of macronutrients; i.e., the diets were isonitrogenous, isolipidic and isoenergetic (Table 1), but with different carbohydrate and protein sources. The control diet comprised of 19% corn starch and 41% fishmeal, and the experimental diet of 8% corn starch, 36% fishmeal and 15% *P. palmata* (hereafter *Palmaria*) meal (Table 1). While protein quality is similar between the diets, the carbohydrates are different because *Palmaria* consists mainly of mono- and polysaccharides, i.e., low-digestible carbohydrates (i.e., high in neutral detergent fibre) (Table 1). Neither feed intake nor growth performance

**Table 1** Atlantic salmon (*Salmo salar*) diet composition and proximate composition for both control and *Palmaria palmata* inclusion experiment modified after *Wan et al. (2016)*.

|  | Control | *Palmaria* inclusion |
|---|---|---|
| *Diet formulation; %* | | |
| Fishmeal[a] | 40.74 | 35.75 |
| Fish oil[a] | 20.00 | 20.41 |
| *Palmaria palmata* | – | 15.00 |
| Extruded corn starch[b] (high-digestible carbohydrates) | 18.76 | 8.34 |
| Wheat gluten[c] | 9.00 | 9.00 |
| Pea protein concentrate[c] | 9.00 | 9.00 |
| Mineral & vitamin premix[d] | 2.00 | 2.00 |
| Antioxidant[e] | 0.50 | 0.50 |
| *Proximate composition[f];* | | |
| Moisture, % | 6.3 | 6.6 |
| Crude protein, % | 40.7 | 40.6 |
| Crude lipid, % | 25.1 | 25.5 |
| Ash, % | 8.3 | 9.9 |
| Gross energy, MJ Kg$^{-1}$ | 26.2 | 26.1 |

**Notes.**

*Palmaria* proximate composition: moisture 9%; crude protein 22%; crude lipid 1%, ash 25% and gross energy 15 MJ kg$^{-1}$. And the estimated neutral detergent fibre, i.e., carbohydrates that resist to digestion and absorption, from *Palmaria* is 43%.

[a] United fish products Ltd., Donegal, Ireland.

[b] Laboratory grade, Sigma–Aldrich Company Ltd., UK.

[c] Purified feed ingredients, Roquette, France.

[d] Premier nutrition products Ltd., UK. (Manufacturers analysis: Ca-12.09%, Ash-78.71%, Na-8.86%, Vitamin A-1.0 $\mu$g kg$^{-1}$, Vitamin D3 0.10%, Vitamin-E 7.0 g kg$^{-1}$, Cu-250 mg kg$^{-1}$, Mg 15.6 g kg$^{-1}$ and P 5.2 g kg$^{-1}$).

[e] Barox plus liquid, Kemin Europa N.V., Belgium.

[f] $n = 4$.

differed significantly between the two diet groups (*Wan et al., 2016*). We hypothesize that substitution of high- with low-digestible carbohydrates, i.e., substitution of corn starch with *Palmaria*, will decrease overall carbohydrate digestibility and availability, which in turn affect how metabolic intermediates are routed in the glycolysis pathway. The glycolytic AA are synthesized from two intermediates in glycolysis: 3-phosphoglyerate (Gly and Ser) and pyruvate (Ala) (*Moran et al., 2012*). We focus on the glycolytic AA because previous studies have shown that they are particularly sensitive biomarkers of carbohydrate and lipid sourcing and metabolism. These macronutrients also serve as the primary energy sources, and increased reliance of carbohydrates for energy may lead to a corresponding decrease in the demand of lipids for energy. Thus, shifts metabolic routing in the glycolytic pathway is likely to affect $\delta^{13}$C values of glycolytic AA because lipids are generally 5–8% depleted in $^{13}$C compared to carbohydrates and protein. By decreasing the fraction of high-digestible carbohydrates, we posit that downstream products directly associated with the glycolysis pathway would become more $^{13}$C enriched in the *Palmaria* than control treatment (Control) because lipid derived intermediates would be routed to energy rather than tissue formation. We also analyzed the EAA to test whether EAA assimilation efficiency and protein preference would be similar for terrestrial (i.e., wheat gluten, pea protein concentrate) and marine (i.e., fishmeal, *Palmaria*) derived protein sources. Finally,

to obtain the supporting information of the fate of dietary macromolecules, we also measured bulk carbon and nitrogen isotopes in the diet and fish.

## MATERIAL AND METHODS

### Source of salmon fillet and diet protein ingredients

The feeding trial was conducted at the Carna Research Station, Ryan Institute, NUI Galway, Ireland, where inclusion of red (*P. palmata*) macroalga was compared to the Control group fed on a basal diet formulation. The salmon were fed on the experimental diets for 14 weeks to ensure that at least half of the muscle carbon pool would reflect the new experimental diets (*Jardine et al., 2004*). At the end of trials, fish were euthanized by concussion, and pithing of the cranium. Feeding trial was carried out under the oversight of National University of Ireland Galway's Animal Care Research Ethics. Details of the experimental design and diet compositions for the feeding study was previously published in (*Wan et al., 2016*). In short, the original feeding experiments were a factorial $4 \times 1$ design with varying macroalgae percentage (0%, 5%, 10% and 15%) as the main factor. For this study, we only included salmon from the Control group and 15% macroalgae inclusion group. For each treatment, we analyzed three fish from each of three different tanks. The Control group were fed on ∼40% of fishmeal as the main source of protein. Both diets were isonitrogenous (40%), isolipidic (25%) and isoenergetic (26 MJ kg$^{-1}$). The experimental group were fed alternative diet substituted with 15% of dried macroalgae into the diet by decreasing fishmeal by 5% and extruded corn starch by 10%, while the compositions of other ingredients remained unchanged (*Wan et al., 2016*). Both Control and experimental diets also comprised two commonly used terrestrial protein sources: pea protein concentrate and wheat gluten meal at the same inclusion rate (9%). No AA were added to the diet formula. A summarized description of the diet information and proximate composition for both Control and *P. palmata* inclusion experiment is presented in Table 1 (modified after *Wan et al., 2016*). The effect of different experimental diets on growth performance, morphometric indices, and feed parameters (i.e., weight gain, feed conversion ratio, and specific growth rate etc.) was monitored throughout the experiment to ensure satisfactory growth rate (Fig. 1). In short, there were no significant differences in the growth parameters (final weight and weight gain) and growth performance indices between the Control and the *Palmaria* inclusion diets (*Wan et al., 2016*). Fillet muscle samples were collected after the fish was euthanized by a sharp blow to cranium and followed by pithing of the brain. Mixed sex salmon smolt were sourced from Derrylea Holdings Ltd. (Lough Fee, Connemara, Ireland) and the detailed information of fish care was described in (*Wan et al., 2016*).

### Stable isotope analyses

We measured the carbon stable isotope ratios of the individual amino acids ($\delta^{13}C_{AA}$) of salmon and diet at Leibniz-Laboratory for Radiometric Dating and Stable Isotope Research in Kiel, Germany. Freeze-dried fish fillet were homogenized and approximately 4 mg of each sample were analyzed for compound specific stable isotope analyses (*Wang et al., 2018*). For the diet samples, three replicate samples of each compound diet

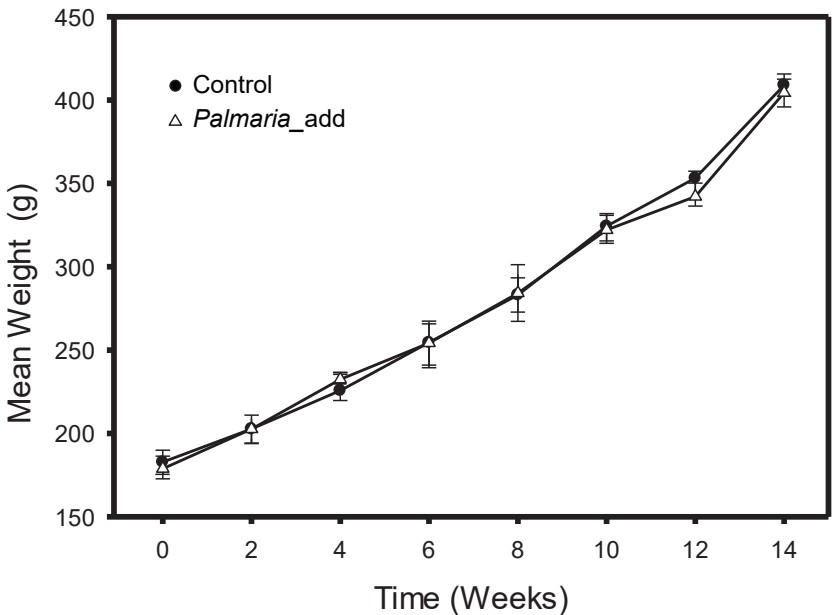

**Figure 1** Growth weight curve (mean ± SD, $n = 3$) of Atlantic salmon smolt for the both the Control and *Palmaria palmata* inclusion treatments during the fourteen-week feeding trial.

and individual dietary components (each homogenized, ∼10 mg each) were analyzed for CSIA on a Gas Chromatogram (GC) connected to a MAT 253 (Thermo-Fisher Scientific (Bremen) GmbH, Bremen, Germany) Isotope Ratio Mass Spectrometer (IRMS). The detailed procedure for AA hydrolyses and derivatization, carbon correction and data calibration as well as the GC temperature setting is described in *Wang et al. (2018)*. In short, each sample was hydrolyzed in 1mL of 6 N HCl at 110 °C in a heating block for 20 h. To remove lipophilic compounds before derivatization, we vortexed the hydrolyzed samples with 2 ml n-hexane/DCM (6:5, *v/v*) for 30 s and subsequently discarded the solvent from the aqueous phase. The AA were derivatized to *N*-acetyl methyl esters following the protocols by *Larsen et al. (2013)* and *Corr, Berstan & Evershed (2007)*. We also derivatized and analyzed a mixture of pure AA with known $\delta^{13}C$ values under the same conditions as the samples to calculate correction factors specific to each AA to account for carbon addition and fractionation during derivatization (*Larsen et al., 2013*).

Isotope data are expressed in delta ($\delta$) notation:

$$\delta^i E_{\text{sample}} = \frac{\left(\frac{iE}{jE}\right)_{\text{sample}} - \left(\frac{iE}{jE}\right)_{\text{ref}}}{\left(\frac{iE}{jE}\right)_{\text{Ref}}}.$$

For the certain element E, the ratio of heavy (i) to light (j) isotope are measured in both sample and references (*Coplen & Shrestha, 2016*). Since isotope values are small numbers, they are multiplied by 1000 and expressed as per mil (‰). Calibration of our internal standard AA-mixture was carried out against the known $\delta^{13}C$ values of A4 mixture (available from A. Schimmelmann, Biogeochemical Laboratories, Indiana

University, USA). The isotope ratios are expressed relative to international standards Vienna Pee Dee Belemnite (VPDB) for carbon. In regard to analytical uncertainty, the average reproducibility for the internal reference standard nor-leucine (Nle) was 0.3‰ ($n = 3$ for each batch) and the in-house AA standards ranged from 0.2‰ for Pro to 0.6‰ for Ala ($n = 4$–7 for each batch). We were able to analyze the following 15 AA: NEAA; alanine (Ala), asparagine/aspartic acid (Asx), glutamine/glutamic acid (Glx), glycine (Gly), proline (Pro), tyrosine (Tyr) and serine (Ser), and EAA; histidine (His), isoleucine (Ile), leucine (Leu), lysine (Lys), methionine (Met), phenylalanine (Phe), threonine (Thr), and valine (Val). See Fig. S1 online for a chromatogram of salmon muscle generated from the GC-Combustion-Isotope Ratio Mass Spectrometer (GC-C-IRMS).

Duplicates of compound diets and individual protein ingredients were analyzed for bulk carbon and nitrogen at the Stable Isotope Facility of the Experimental Ecology Group, GEOMAR, Kiel, Germany. Approximately 100 µg dry mass of each sample was weighed into tin capsules and analyzed on a customised elemental analyzer (EA 1110, Fisons Instruments, Milan, Italy) connected to a ThermoElectron DeltaPlus Advantage IRMS (Hansen & Sommer, 2007). The isotope ratios are expressed relative to international standards; VPDB for carbon and atmospheric air for nitrogen. For the detailed description of the calibration of the data, see description previously in Wang et al. (2018). In short, calibration of internal standard was carried out against certified reference material (IAEA-N1, IAEA-N2, IAEA-N3 for $\delta^{15}N$ and IAEA-CH-3, IAEA-CH6, IAEA-CH-7 for $\delta^{13}C$). Internal standard conducted for bulk $\delta^{13}C$ and $\delta^{15}N$ analyses during the sample sequence yielded $1\sigma = 0.2$‰ and 0.2‰, respectively. Because lipids have more negative $\delta^{13}C$ values than proteins, it is important to account for the large variability in lipid content, which can affect the $\delta^{13}C$ values of bulk tissue and lead to false interpretation. Thus, the negative $\delta^{13}C$ values are commonly corrected by extracting lipids from samples prior to isotope analyses, or applying a mathematical correction after isotope analyses based on sample C:N ratios (Logan et al., 2008; Logan & Lutcavage, 2008). We opted for the latter and applied posterior correction to the $\delta^{13}C$ values if the fishmeal samples C:N ratio was greater than 3.5, following Logan et al. (2008). Given that the fishmeal is non-tissue specific and of marine origins, we used the all-tissue correction parameters for marine fish (Logan et al., 2008). A similar posterior correction was also applied to fish muscle as previously described and published in Wang et al. (2018).

## Statistical analyses

All statistical analyses were performed using R version 3.4.3 (R Core Team, 2017) (Supplementary Information). All values in the text are given as mean and its corresponding standard deviation (SD). For each tank, the mean $\delta^{13}C_{AA}$ values of fish are based on triplicate fish analyses. Isotope offset between fish muscle (F) and diet ($\Delta^{13}C_{F-D}$) were calculated for all amino acid and bulk isotopes of each treatment as $\Delta^{13}C_{F-D} = \delta^{13}C_F - \delta^{13}C_D$, where the $\delta^{13}C_F$ and $\delta^{13}C_D$ represent the $\delta^{13}C$ values of the fish (i.e., Atlantic salmon) and diet (both compound diet and individual protein ingredients), respectively. Assuming independent variables, the error term for $\Delta^{13}C_{F-D}$ are propagated according to $\sqrt{SD_F^2 + SD_D^2}$ (Ku, 1966). We used univariate Analysis of Variance (ANOVA)

performed on the output from Multivariate Analysis of Variance (MANOVA) to access which dependent variables ($\delta^{13}C_{AA}$ values) are significantly different between groups (R function: summary.aov). The $P$-adjusted values from this ANOVA decomposition test was obtained using *p. adjust* function with FRD method (*Benjamini & Hochberg, 1995*). We did not perform student $t$-test for comparing the bulk isotope values of two compound diet as we only have duplicate runs for each diet. The student $t$-tests were performed to compare the mean EAA $\Delta^{13}C_{F-D}$ values of both treatments. Unless otherwise stated, statistical significance is assessed at $P < 0.01$. Due to our small sample numbers, we chose a 0.01 significance level ($\alpha$) to decrease the probability of Type I error. The statistical tests results are presented in the Supplementary Information.

## RESULTS

### Amino acid and bulk carbon isotope values of compound diets and protein ingredients

Among the four different dietary ingredients, $\delta^{13}C_{AA}$ values of fishmeal were the highest for all AA except for Lys and Met, which were slightly lower than those of *Palmaria* by 1–2‰. In general, $\delta^{13}C$ values (both AA and bulk) of marine sourced proteins (i.e., fishmeal and *Palmaria*) were much higher than the terrestrial protein sources (Fig. 2 and Tables S1 and S2) with the exception that the Gly $\delta^{13}C$ values of *Palmaria* is slightly lower than that of wheat gluten but higher than pea protein. Accordingly, $\delta^{13}C_{AA}$ and $\delta^{13}C_{bulk}$ values of the compound diets are in between those of the marine and terrestrial dietary sources (Fig. 2 and Table S1). For the NEAA, the $\delta^{13}C$ range was widest for Pro, Ser, and Gly (∼13 ‰) and narrowest for Ala and Tyr (∼6 ‰). For the EAA, the range was widest for Met (∼12 ‰) and narrowest for Phe (∼6 ‰). Among all AA in the individual ingredients, Ser, Gly and Thr were the most $^{13}C$-enriched in both treatments, whereas Tyr, Phe, and Leu were the most $^{13}C$-depleted (Fig. 2). In general, $\delta^{13}C_{AA}$ and $\delta^{13}C_{bulk}$ values of marine sourced protein fishmeal and *Palmaria* were more similar except for Ser and Gly (up to 8.9‰ difference). Likewise, individual AA $\delta^{13}C$ and $\delta^{13}C_{bulk}$ values of wheat gluten were similar to those of pea protein concentrate except for Gly and Ile (Fig. 2). Although the mean bulk $\delta^{13}C$ value of the *Palmaria* containing diet was 1.1‰ lower than that of the Control diet, $\delta^{13}C_{bulk}$ values of fish muscle for both Control and *Palmaria* inclusion groups are identical (both $\delta^{13}C = 20.2$‰) (Fig. 2 and Table S1).

### $\delta^{13}C_{NEAA}$ offsets between fish muscle and compound diets

There was a large variability in $^{13}C_{F-D}$ offset ($\Delta^{13}C_{F-D}$) between fish and the compound diet for the individual NEAA. The $\Delta^{13}C_{F-D}$ values of all NEAA except Asx (∼0‰) showed much greater variability, ranging from Ala ($-6.8$‰) to Ser (5.9‰) (Figs. 2A and 2B). Moreover, $\Delta^{13}C_{F-D}$ values of the NEAA were typically higher than their respective compound diet, with the exception of Ala in both treatments and Gly in the Control treatment. Furthermore, $\Delta^{13}C_{F-D}$ values of glycolytic amino acids and Tyr in *Palmaria* treatment were significantly different from those in Control diet treatment (ANOVA, $P$-adjusted $< 0.01$, Table S3). In contrast, there was no difference between two treatments for Krebs cycle AA Glx, Asx and Pro (ANOVA, $P$-adjusted=0.03, 0.153 and 0.25, respectively, Table S4). The NEAA

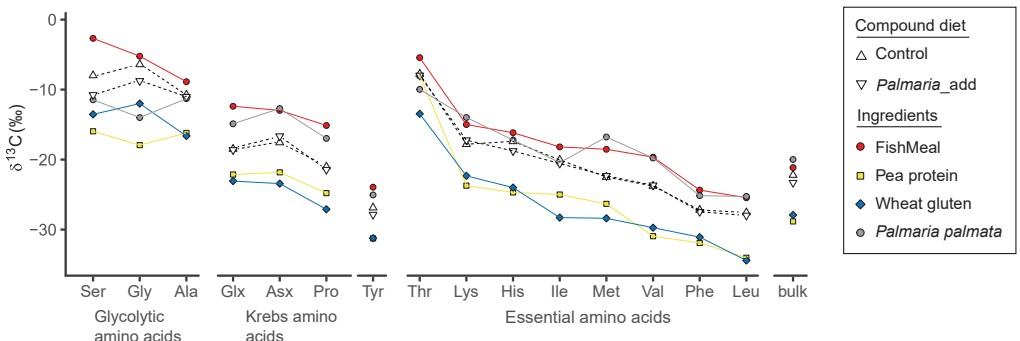

**Figure 2** $\delta^{13}C_{AA}$ and $\delta^{13}C_{bulk}$ values (mean ± SD) of compound diet and individual dietary protein components for both Control and experimental *Palmaria palmata* inclusion treatments. Labels for the compound diets for both Control and experimental fish trials: Control and *Palmaria*_add; The dietary ingredients are fishmeal, pea protein concentrate, wheat gluten, and *P. palmata*. For visual clarity, the amino acids for individual dietary ingredients are connected by lines of different colors.

$\Delta^{13}C_{F-D}$ values in *Palmaria* fed fish were generally larger (e.g., up to 3.4‰ for Gly) than those of the Control fed fish. The patterns observed for $\Delta^{13}C_{F-D}$ values of all NEAA also mirrored the patterns observed in the $\Delta^{13}C_{F-D}$ of bulk samples (Figs. 3A and 3B).

## $\delta^{13}C_{EAA}$ offsets between fish muscle and compound diets

The $\delta^{13}C_{EAA}$ patterns of fish muscle largely to reflect that of the compound diets instead of individual protein components for both treatments (Fig. 4A and Fig. S2). The $\delta^{13}C_{EAA}$ values of the fish muscle and compound diet are almost identical, resulting in a mean offset of 0.34‰ and 0.18‰, respectively for Control and *Palmaria* inclusion group (Figs. 4A and 4B). The mean EAA $\Delta^{13}C_{F-D}$ values of both treatments were not significantly different from 0‰ (one sample $t$-test, $t = 1.90$, $df = 7$, $P = 0.099$ for Control diet; $t = 1.59$, $df = 7$, $P = 0.213$ for *Palmaria* inclusion diet). There was no apparent isotopic offset between salmon muscle (fish) and diets ($\Delta^{13}C_{F-D}$) for EAA (Figs. 4A and 4B) except for Met being $1.3 \pm 0.7$‰ more positive than the compound diet in the Control diet group. Also, $\Delta^{13}C_{F-D}$ values of individual EAA were not significantly different between *Palmaria* inclusion and Control treatment except for His and Met (ANOVA, both $P$-adjusted < 0.01, detailed ANOVA results at Table S4).

## DISCUSSION

Our isotopic results show that Atlantic salmon from the two experiments utilized the macronutrients differently because substitution of high- with low-digestible carbohydrates led to more positive offsets between fish and their diets ($\Delta^{13}C_{F-D}$) of the glycolytic AA, Ser, Gly and Ala. In our study, the starch fraction of the diets (i.e., 19% for Control vs. 8% for *Palmaria*_inclusion experiment) was the greatest variable that influenced $\Delta^{13}C_{F-D}$ because both lipid sources were kept consistent and proteins were functionally similar between the two compound diets (*Wan et al., 2016*). Therefore, we argue that the primary

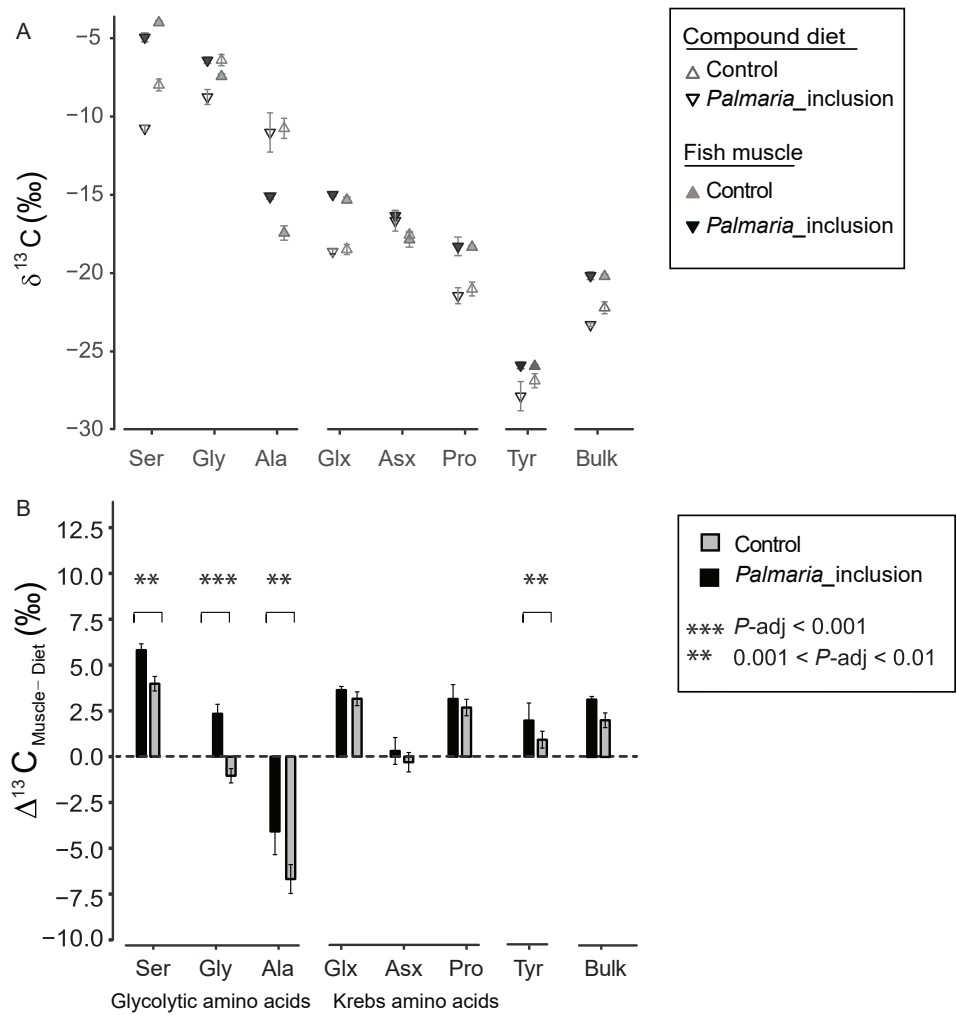

**Figure 3** **Non-essential amino acid and bulk $\delta^{13}$C values (mean ± SD) of fish muscle and compound diets (A) and their offset ($\Delta^{13}C_{F-D}$) (mean ± SD) for both Control and *Palmaria palmata* inclusion experiments (B).** Glycolytic amino acids and Tyr $\Delta^{13}C_{F-D}$ values in *Palmaria* treatment were significantly different from those in Control diet treatment (ANOVA, *P*-adjusted < 0.01, Table S3). In contrast, there was no difference between two treatments for Krebs cycle amino acids Glx, Asx and Pro (ANOVA, *P*-adjusted = 0.03, 0.153 and 0.25, respectively, Table S3). We did not perform statistical comparison between the bulk samples as we only measured bulk isotopes of fish and diet twice each.

cause for the greater discrimination factor of the three glycolytic AA can be ascribed to inclusion of low-digestible carbohydrates.

The more negative bulk $\delta^{13}$C values of the *Palmaria* than Control compound diet (Fig. 3A) can largely be attributed to two factors: (1) Gly and Ser are $^{13}$C depleted by 8–10‰ in *Palmaria* compared to fishmeal proteins (Fig. 2), and (2) bulk *Palmaria* is ca. 10‰ more $^{13}$C depleted (−20‰, this study, Table S1) than corn starch (−10‰, *Tieszen & Fagre, 1993*). Despite the more negative values of these compounds and ingredients in the *Palmaria* diet, the $\Delta^{13}C_{F-D}$ values of bulk and the three glycolytic AA are larger for the *Palmaria* than the Control salmon group (Figs. 3A and 3B), indicating that the intermediates

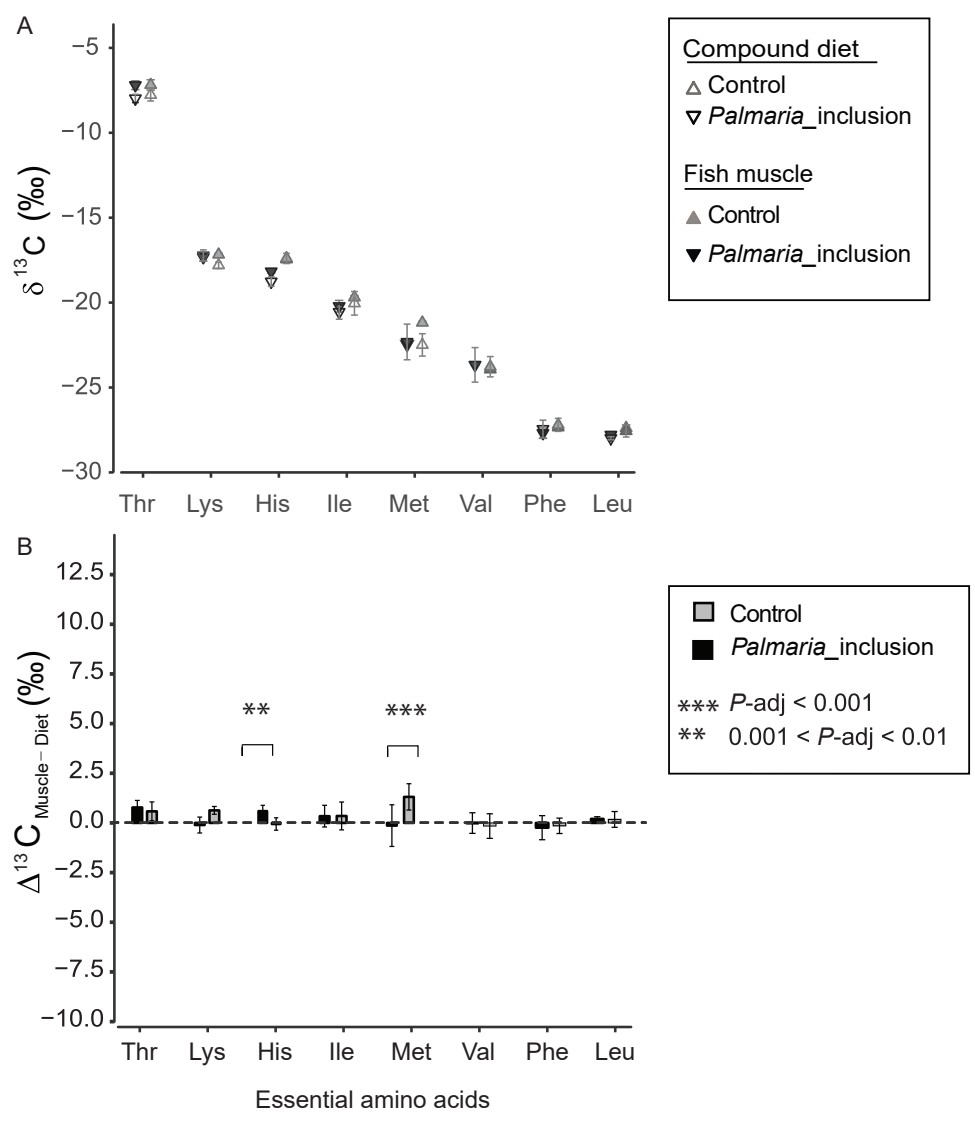

**Figure 4** Essential amino acid $\delta^{13}$C values (mean ± SD) of fish muscle and compound diets (A) and their offset ($\Delta^{13}$C$_{C-D}$) (mean ± SD) (B) for both Control and *Palmaria palmata* inclusion experiments. The mean EAA $\Delta^{13}$C$_{F-D}$ values of both treatments were not significantly different from 0‰ (one sample $t$-test, $t = 1.90$, $df = 7$, $P = 0.10$ for Control diet; $t = 1.59$, $df = 7$, $P = 0.22$ for *Palmaria* inclusion diet). Also, $\Delta^{13}$C$_{F-D}$ values of individual EAA were not significantly different between *Palmaria* inclusion and Control treatment except for His and Met (ANOVA, both $P$-adjusted < 0.01, detailed ANOVA results at Table S4).

used for synthesizing glycolytic AA were more $^{13}$C enriched for *Palmaria*_inclusion than Control salmon (Figs. 3A and 3B). Hence, it is unlikely that utilization of carbohydrates *per se* led to the more positive $\Delta^{13}$C$_{F-D}$ values in the glycolytic AA. Since lipids have $\delta^{13}$C values that are 3–8‰ lower than proteins and carbohydrates (*Cherry et al., 2011*), we posit that lack of easily-digestible starch carbohydrates in *Palmaria* (*Jiao et al., 2012*)

increased the demand for dietary lipids as an energy source and correspondingly decreased the fraction of lipids available for biosynthesizing NEAA intermediates.

A recent study comparing Atlantic salmon fed compound diets with isotopically distinct lipid sources found that lipids are sourced to glycolytic rather than Krebs AA (*Wang et al., 2018*). This result indicates that carbon for acetyl CoA, an upstream precursor of Krebs cycle AA, is sourced more from protein- than lipid-derived building blocks. In our study, we found no $\Delta^{13}C_{F-D}$ difference in Krebs AA between the two treatments further corroborating that Krebs AA are sourced more from proteins than lipid and carbohydrates. It is possible that the decrease in digestible starch content also increased gluconeogenesis, i.e., generation of glycolytic metabolites from AA oxidation. How this pathway affect the discrimination factor for the glycolytic AA in an obligate carnivore is unclear. A study with calorie restricted rats demonstrated that increased AA oxidation led to a $\Delta^{13}C_{F-D}$ decrease in proteins and their constituent NEAA and EAA (*Huneau et al., 2019*). As already mentioned above, we observe a general $\Delta^{13}C_{F-D}$ increase rather than decrease in the *Palmaria* treatment for the NEAA (significant for half of them) and no change for the EAA (Fig. 4 and Table S3). Thus, we find it less likely that *Palmaria* inclusion led to increased AA oxidation. Finally, we rule out the possibility that relative differences in the abundance of dietary AA contributed to the different discrimination factors between the two treatments because the molar balance between the two composite diets are almost identical (Table S5).

Interestingly, $\Delta^{13}C_{F-D}$ values were positive for Ser and Gly and negative for Ala indicating that pyruvate was more $^{13}C$ depleted than 3-phosphoglycerate. Several processes could have contributed to this contrasting $^{13}C$-offset between the two glycolytic intermediates. Pyruvate is a major substrate for oxidative metabolism, and a branching point for synthesis of lactate, fatty acid, alanine and Krebs cycle intermediates (*McCommis & Finck, 2015*). While pyruvate in Atlantic salmon is mainly produced via the glycolytic pathway, it can also be produced by conversion of lactate and oxaloacetate (*Kanehisa et al., 2016*). How these multiple pathways contribute to isotopic discrimination of pyruvate relative to 3-phosphoglycerate is exceedingly complex because isotopic discrimination depends on the intrinsic fractionation of a reaction and the relative flow of intermediates through that reaction pathway. For Atlantic salmon, glycolysis appears to be the main pathway for synthesizing both pyruvate and 3-phosphoglycerate because change in the starch fraction of the diets from 19% to 8% and consequent change in lipid metabolism led to similar directional change in the isotope discrimination factor.

Enhanced gut microbial activity in the *Palmaria* treatment perhaps offers an alternative explanation for the more positive $\Delta^{13}C_{F-D}$ values of the glycolytic AA in the *Palmaria* treatment. Such activity would lead to increased loss of $^{13}C$ depleted $CO_2$ resulting in more $^{13}C$ enriched metabolic intermediates for NEAA biosynthesis (*Larsen et al., 2016*). This alternative possibility rests on the premise that the greater viscosity and lower digestibility of the seaweed carbohydrates compared to corn starch slowed down gut transit time and increased microbial digestion of dietary carbohydrates, i.e., seaweed inclusion changed the salmons' gut microbiome to facilitate carbohydrate digestion and metabolism (*Gajardo et al., 2016*). However, we find the putative effect of microbial activity on the discrimination factor less likely because *Palmaria* inclusion did not appear to affect growth performance

(*Wan et al., 2016*) indicating that the fish did not lack energy or metabolic intermediates. Finally, the offset in $\delta^{15}N$ values between fish muscle and diet between the two groups ($\Delta^{15}N_{F-D}$ values) are the same further suggesting no enhanced microbial activity in the *Palmaria*_inclusion experiment (Table S1).

Unlike the other NEAA, the $\Delta^{13}C_{F-D}$ of Asx is ~0‰ for both the Control and *Palmaria* salmon, which is an unusual finding in fish feeding studies (*McMahon et al., 2010*; *Newsome et al., 2011*; *Whiteman et al., 2018*). The virtually identical $\delta^{13}C_{Asx}$ values between salmon and their compound diets does not necessarily mean that Asx was routed exclusively from dietary sources to muscle tissue because Atlantic salmon do produce enzymes for catalysing the interconversion of oxaloacetate and glutamate to aspartate and $\alpha$-ketoglutarate (*Kanehisa et al., 2016*). If salmon indeed synthesized Asx *de novo*, $^{13}C$ fractionation during Asx synthesis coincidently led to similar $\delta^{13}C$ values between diet and *de novo* synthesized Asx. Further feeding studies varying all three macronutrients (i.e., protein, lipid and carbohydrates) would be needed to explain the isotope discrimination factor of Asx.

We also compared $\Delta^{13}C_{F-D}$ values of EAA between Atlantic salmon and their protein sources to investigate EAA assimilation of terrestrial and marine protein sources. For the EAA, $\delta^{13}C_{EAA}$ values between fish muscle and their compound diets are mostly identical in both treatments except for Met (Figs. 4A and 4B). This finding corroborates with previous findings that EAA is passed on from dietary sources to fish without alternation of their carbon skeletons (*McMahon et al., 2010*). Our study indicates that there was no preferential EAA assimilation of any of the three protein source in the compound diets because $\delta^{13}C_{EAA}$ values in salmon were similar between the two treatments and to the compound diets. This finding was expected given that feed conversion ratios (FCR for Control, $1.28 \pm 0.10$; and for *Palmaria* diet, $1.32 \pm 0.03$) during the feeding trial were comparable across the macroalgal inclusion diet and Control diet (*Wan et al., 2016*). Our near zero $\Delta^{13}C_{F-D}$ values for all EAA in both feeding experiments rule out, unsurprisingly, EAA supplementation by gut microbes to host (*McMahon et al., 2010*).

Both protein digestibility and availability in commercial aquafeeds are designed to ensure maximum growth and feed utilization. This was also the case for the diets in our feeding trial (*Wan et al., 2016*); hence, both our diets had almost identical AA composition (see Table S5). It is also evident from the AA composition data that there are apparent AA mismatches between salmon and their diets. This raises the question whether these mismatches correlate with NEAA discrimination factors, because dietary deficiencies could lead to higher synthesis rates of particular NEAA to meet metabolic demands. We did not find such correlations ($R^2 = 0.0003$ for Control and $R^2 = 0.08$ for *Palmaria* diet, respectively), which underlines that discrimination factors are determined by multiple factors. Our study highlights that one of those factors is metabolic utilization of dietary macromolecules; in our case how carbohydrate availability presumably affect lipid utilization. Other factors that could affect metabolic demands are infection, recoveries from injury, physical activities and digestive processes (*Dunstan et al., 2019*). In addition, proteomics studies show that synthesis and degradation rates of individual proteins in fish vary greatly (*Doherty et al.,*

*2012*). Since the AA composition vary between proteins, this is another factor that may lead to a disproportionally greater turnover of certain AA than others.

One potential bias in using $\delta^{13}C_{AA}$ to assess dietary routing in salmon is whether the AA yield from acid protein hydrolysis in the laboratory is comparable to the yield from the salmons' digestive system. Given that we have an analytical uncertainty of approximately 0.5‰ and a dynamic range between terrestrial and marine derived AA of 7–12‰, our results suggests that this bias is very small because no EAA except Met have significantly different $\delta^{13}C_{EAA}$ values between salmon muscle tissues and compound diets in the Control group. In this case, Met of fish muscle has slightly higher $\delta^{13}C$ values than their compound diet by $1.3 \pm 0.7$‰. This result may be caused by methionine loss during acid hydrolysis (*Jennings & Lewis, 1969*) or analytical uncertainty due to the low abundance of Met (Fig. S1). Alternatively, $^{13}C$ fractionation of methionine may occur when it is used for other purposes than protein synthesis, e.g., for synthesizing cysteine used for producing pancreatic proteases (*Holm, Fossum & Eide, 1973*).

## CONCLUSIONS

Our previous study (*Wang et al., 2018*) demonstrated that $\delta^{13}C$ values of the glycolytic AA are sensitive markers of lipid origins (i.e., terrestrial vs. marine). By comparing treatments where corn meal was replaced by macroalga, we show that glycolytic AA are sensitive to the dietary carbohydrate sources and digestibility. The most parsimonious explanation for the more positive $\Delta^{13}C_{F-D}$ values of the glycolytic AA in the *Palmaria* treatment is a decreased sourcing of lipid derived intermediates to AA synthesis because *Palmaria* carbohydrates are more $^{13}C$ depleted than corn starch. While more feeding studies are warranted for understanding metabolic routing of macronutrients, our findings show how characterization of $\delta^{13}C_{AA}$ variability can be used to trace relative contributions of dietary carbohydrates, proteins and lipids for *de novo* NEAA biosynthesis for carnivorous bony fish raised under normal feeding and husbandry conditions. Furthermore, our results indicate $\delta^{13}C_{AA}$ analysis can provide supplementary information on the gut microbiome's role in carbohydrate digestion and metabolism. The wider adoption of compound specific isotope analysis, particularly for AA, can greatly improve our understanding of nutrient utilization during the fish growth of different aquaculture species and life stages (*Le Vay & Gamboa-Delgado, 2011*; *Newsome et al., 2011*; *Whiteman et al., 2018*). Feeding studies such as ours also help in validating assumptions and limitations on how to interpret $\delta^{13}C_{NEAA}$ values in ecological studies.

## ACKNOWLEDGEMENTS

The authors would like to thank Robert Priester for the laboratory assistance and Karsten Gramenz and Dr. Nils Andersen at the Leibniz Laboratory, Kiel for technical assistance. We are grateful for Prof. Yoshito Chikaraishi and one anonymous reviewer for their valuable comments, which improved the manuscript.

### Funding

This work was supported by the Cluster of Excellence 80 ''The Future Ocean''. The ''Future Ocean'' is funded within the framework of the Excellence Initiative by the Deutsche Forschungsgemeinschaft (DFG) on behalf of the German federal and state governments. Yiming V. Wang was supported by the German Federal Ministry of Education and Research (BMBF) [grant No. 03F0722A] during 2017–2018. Thomas Larsen is supported by BMBF [grant No. 07F00805A]. Alex H.L. Wan was supported by Grant-Aid [Agreement No. MFFRI/07/01] under the Sea Change Strategy with the support of the Marine Institute and also National Development Plan 2007–2013 grant to the Department of Agriculture, Food and the Marine, Ireland. The funders had no role in study design, data collection and analysis, decision to publish, or preparation of the manuscript.

### Grant Disclosures

The following grant information was disclosed by the authors:
Cluster of Excellence 80 ''The Future Ocean''.
Excellence Initiative by the Deutsche Forschungsgemeinschaft (DFG) on behalf of the German federal and state governments.
German Federal Ministry of Education and Research (BMBF): 03F0722A.
BMBF: 07F00805A.
Department of Agriculture, Food and the Marine, Ireland: MFFRI/07/01.

### Competing Interests

The authors declare there are no competing interests.

### Author Contributions

- Yiming V. Wang and Thomas Larsen conceived and designed the experiments, analyzed the data, contributed reagents/materials/analysis tools, prepared figures and/or tables, authored or reviewed drafts of the paper, approved the final draft.
- Alex H.L. Wan performed the experiments, analyzed the data, contributed reagents/materials/analysis tools, prepared figures and/or tables, approved the final draft.
- Åshild Krogdahl approved the final draft, editorial view and critical comments and discussion on Atlantic salmon physiology.
- Mark Johnson performed the experiments, contributed reagents/materials/analysis tools, approved the final draft.

### Animal Ethics

The following information was supplied relating to ethical approvals (i.e., approving body and any reference numbers):

NUI Galway Animal Care Research Ethics Committee is the NUI Galway's governing body in Institutional Animal Care and Use. The fish feeding trial reported in the manuscript does not require Animal Care Research Ethics Committee's approval as it did not
conduct any procedures on the animals while they were alive, e.g., tissue harvest. This is a conventional feeding trial and only comprise conventional animal husbandry (feeding, growing and weighing) and were euthanised by trained individuals using approved methods under Irish/EU regulation, e.g., i.e., concussion to the cranium and pithing of the brain.

## Data Availability

The raw data is available at GitHub: https://github.com/alsjmonsoon/Salmon-Feeding-Study. The code is available as Supplemental File.

## Supplemental Information

Supplemental information for this article can be found online at http://dx.doi.org/10.7717/peerj.7701#supplemental-information.

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
