# Peer review of "C values of glycolytic amino acids as indicators of carbohydrate utilization in carnivorous fish"

_PeerJ, doi:10.7717/peerj.7701_

## Round 0.1 · original submission · Major Revisions

Both reviewers found this to be an interesting study, with potential to be an important contribution to the literature. They also provide a number of questions and comments to help improve the manuscript, particularly in regards to how the data are interpreted.

I have two very minor review-level comments of my own: (1) on the x-axis of Fig. 2 I believe "none" should be "non" and (2) delta-13C values cannot be depleted or enriched (see https://digitalrepository.unm.edu/cgi/viewcontent.cgi?filename=2&article=1000&context=unm_oer&type=additional), so please adjust the wording on lines 140, 145, and perhaps elsewhere.

Reviewer 1 ·

Basic reporting

This paper provides a good data for effect of dietary carbohydrate on the d13 C values of amino acids in Atlantic salmon using compound specific stable carbon isotope analysis. In this study, Atlantic salmon were reared either on a control diet with high carbohydrate (corn starch) or on an experimental diet with low carbohydrate (red seaweed, Palmaria). The control diet was a mixture of fish and pea protein (58%), lipids (20%), carbohydrate: wheat gluten (9%) and corn starch (18.7%). The experimental diet added a low carbohydrate ingredient, Palmaria in all the same composition without corn starch. Authors found that there is a wide difference in d13 values of glycolytic amino acids (Ser, Gly and Ala) from diet to body tissues. They suggested that glycolytic amino acids are associated with dietary carbohydrates.

Although the paper contains potentially interesting information, my main concern with this paper is that authors need to clarify the composition of main three dietary nutrients (proteins, lipids, carbohydrates) within high carbohydrate diet (Control) and low carbohydrate diet (Experiment). Furthermore, Authors need to clarify the d13 C values of each component. For example, although both corn starch and wheat gluten are carbohydrate, d13C values of corn starch are approximately -9 to 11‰ and wheat gluten is around -25 to -30‰. Average of both mixture is around -20‰. Palmaris is composed of 22% protein and no carbohydrate and d13 values of Palmaris is -20‰. Both control and palmaris added are -20‰ but there is difference in each component. Furthermore, there is other influence of lipids on both Control and Experiment. Authors need to explain how d13C values of lipids are consistent both two diets and then argue that d13C values of glycolytic amino acids came from Palmaria carbohydrate, not from Palmaria lipids and fish oil.

Other recommending on this paper is that it is much better to do more focusing on effect of carbohydrate on glycolytic amino acid syntheses, even though all data came from amino acids. This paper is trying to explain three nutrients (amino acids, carbohydrate and lipids) on one experiment and it is quite confusing. So I suggest it needs to rewrite on each paragraph focusing on the influence of carbohydrate on d13C values of amino acids. Several NEAAs also are strongly related to lipids when there is less amount of proteins in diets. Authors need to explain more why Ser, Gly and Ala was significantly affected by the isotopic values of energy components (carbohydrates) of diet. Alanine is formed by the addition of an amino group to pyruvate, a precursor of glucose. Glycine is synthesized from serine, which is synthesized from 3-phosphoglycerate.

Additionally, authors tried to put all NEAA and EAA data on Figures, but I suggest that all EAA results (no difference) should be sent back to supplementary information and put only the NEAA data on figures to make clear on the main finding.

Experimental design

Experimental design on this paper is based on feeding experiment to test the effect of seaweed carbohydrate on amino acids on fish. Authors compared Atlantic salmon with and without low carbohydrate seaweed Palmaria. All data were well presented and results are interesting for scientific community. However, authors need to do more works on the composition of each diet (control and seaweed inclusion) based on the macronutrients (protein, lipid, carbohydrate) and require more explanation on carbohydrate and glycolytic amino acids.

Validity of the findings

All underlying data were provided well and Conclusions are well stated.

Additional comments

Detailed comments

Line 98-100: Authors mentioned Atlantic salmon and metabolic fate of plant carbohydrate. Plant has protein, lipid and carbohydrate. I think authors need to explain why plant carbohydrate is important in carnivorous fish instead of plant protein or plant lipid. Authors mention Palmaria is low carbohydrate and high protein and it is confusing.

Line 224-225: Salmon muscle contains a lot of lipids compared with other species. Without extracting lipids from salmon muscle tissues, d13C values of amino acids would be biased due to lipid effect. Authors need to explain why the lipid correction was used on muscle tissues and how it is correct. There is increased demand for dietary lipid as an energy source when no carbohydrate in Palmaria. And the sentence “to affect on d15N values” is not relevant and focusing on d13C values and please delete.

Line 255-256: This sentence is not correct. D13C values of marine sourced proteins were higher than the terrestrial protein sources especially Gly is highest in marine proteins and it is used for detecting the consumption of marine food in human diets. Please correct.

Line 272-273: Authors need to explain why Ala is negative and Ser and Gly is positive in d13C values offset between fish and diets on Discussion part. It suggests there is different pathway on synthesis on both negative alanine and positive serine or glycine. Please focus on three glycolytic amino acids along with corn starch or non-starch carbohydrate.

Line 303-304: Authors need to mention this corn starch in Intro part. I suggest authors need to explain how corn starch is different from non-starch polysaccharides.


Table 1: This table should have percentage of protein, lipid, carbohydrate and include each carbon stable isotope values from Control and Experiment diet. Please add three nutrients and carbon stable isotope values of three nutrients. Especially, please add composition of carbohydrate in Palmira.

Figure 2: I suggest that only B is enough for showing the results and better to delete A or delete EAA data on Figure 2 and simplify the figure.

Figure 3: Figure 3 is very confusing and difficult to figure out the difference between control and Palmaria inclusion. Please use only compound diet and Palmaria focusing on difference.

·

Basic reporting

no comment

Experimental design

See 'General comments for the author'

Validity of the findings

See 'General comments for the author'

Additional comments

Dear Authors,

This letter constitutes my review of the manuscript titled “13C values of glycolytic amino acids as indicators of carbohydrate utilization in carnivorous fish” by Yiming V. Wang et al., for PeerJ.

In this study, the Authors investigated the effects of high- and low-digestible carbohydrates in diets to the carbon isotopic composition of amino acids in Atlantic salmon. I feel that the data reported are novel and may be valuable in that they impact how we can understand carbon isotopic composition of amino acids in organisms. However, I feel that the interpretation/explanation of the data observed is insufficient in this manuscript, because I have great concerns (which are described below) in several places in this manuscript. I therefore recommend ‘major revision’ to the handing editor of PeerJ. Sorry for negative comments, but I will be grateful if the Authors positively receive my comments, and if my comments are useful to enhance the value of the revised manuscript.

1. From this manuscript, I did not understand why the cap-delta 13C (muscle vs. diet) values are significantly large or small among non-essential amino acids such as Ser, Gly, Ala, Glx, and Pro, but negligible for Asx. It will be better if the Authors can explain specific mechanism to make such variation in the cap-delta 13C values among non-essential amino acids.
2. Also I did not understand why the variable delta-13C value are related to biosynthetic activity of non-essential amino acids. I think that abundance of non-essential amino acids (except for Alx) should be significantly increased from compared to that of essential amino acids, if the biosynthetic activity of non-essential amino acids is significantly large enough to change in the delta-13C value between salmon and diets for the amino acids. It will be better if the Authors show change in the molar balance (i.e., increase the of non-essential amino acid abundance, except for Alx) of amino acids from diets to salmon.
3. I am very much doubtful of digestibility of corn starch in the Atlantic salmon. I think that the salmon have no chance to feed on any starch in wild environments, and therefore that they have little or no digestibility of corn starch. Moreover, in general, for human, we can easily digest cooked (e.g., boiled) starch, but hardly digest raw starch. It will be better if the Authors show digestibility of ‘raw’ corn starch for salmon that used in this study. Probably, reference literature(s) for line 303-304 is(are) required.
4. In this manuscript, a little information is available for the controlled feeding experiments of the salmon. I think that, for example, initial size of the salmon before experiments and change in the biomass size (e.g., length, weight, etc.) during the experiments are at least required. These data will be very useful for readers of this manuscript, to recognize that the 14 weeks are enough in this experiments. Perhaps, it will be better if the Authors show the growth carves of salmon during the trials in this experiment.

Please do not hesitate to contact me if you have any questions about my review.

Sincerely yours,

July 6, 2019
Yoshito Chikaraishi
ychikaraishi@lowtem.hokudai.ac.jp

---

## Round 0.2 · Minor Revisions

The manuscript has been re-reviewed by Dr. Chikaraishi. He requests additional explanation regarding the molar balance of amino acids and I would appreciate it if the authors could address this concern.

·

Basic reporting

no comment

Experimental design

no comment

Validity of the findings

no comment

Additional comments

I have four concerns on the previous manuscript, and I recognize that the Authors have made revision sufficiently against the review comments (1), (3), and (4), but not the comment (2) about change in the molar balance of amino acids from diets to salmon.

I appreciate that the Authors added a supplemental table (Table S5) for showing AA compositions for diets and salmon. However, my original concern is increasing when I saw them. The Authors have explained that the variable delta-13C value are related to biosynthetic activity of non-essential amino acids (except for Alx). If this explanation is true, I think that abundance of non-essential amino acids (except for Alx) should be significantly increased compared to that of essential amino acids, because the biosynthetic activity of non-essential amino acids is significantly large enough to change in the delta-13C value while the biosynthetic activity of essential amino acids is zero. However, in Table S5, I did not find such increase of non-essential amino acid concentration in the AA composition from diets to salmon. Thus I think that ‘the Authors’ explanation for the delta-13C value (i.e., large biosynthetic activity of non-essential amino acids)’ and ‘observation for the AA composition (i.e., little increase of non-essential amino acid concentration)’ may be contradictory to each other.
I think that this concern is not main point of this study, but I would like to see potential/possible explanation for this contradiction.

Sincerely yours,
August 10, 2019
Yoshito Chikaraishi
ychikaraishi@lowtem.hokudai.ac.jp

---

## Round 0.3 · accepted · Accept

Thanks for your thorough revisions to the manuscript.

#